# *Trans-LoRA*: towards
# data-free Transferable Parameter Efficient Finetuning

**Runqian Wang**[*†‡]
raywang4@mit.edu

**Soumya Ghosh**[†]
ghoshso@us.ibm.com

**David Cox**[†]
david.d.cox@ibm.com

**Diego Antognini**[‡]
diego.antognini@ibm.com

**Aude Oliva**[*]
oliva@mit.edu

**Rogerio Feris**[†]
rsferis@us.ibm.com

**Leonid Karlinsky**[†]
leonidka@ibm.com

## Abstract

Low-rank adapters (LoRA) and their variants are popular parameter-efficient fine-tuning (PEFT) techniques that closely match full model fine-tune performance while requiring only a small number of additional parameters. These additional LoRA parameters are specific to the base model being adapted. When the base model needs to be deprecated and replaced with a new one, all the associated LoRA modules need to be re-trained. Such re-training requires access to the data used to train the LoRA for the original base model. This is especially problematic for commercial cloud applications where the LoRA modules and the base models are hosted by service providers who may not be allowed to host proprietary client task data. To address this challenge, we propose *Trans-LoRA*— a novel method for lossless, nearly data-free transfer of LoRAs across base models. Our approach relies on synthetic data to transfer LoRA modules. Using large language models, we design a synthetic data generator to approximate the data-generating process of the *observed* task data subset. Training on the resulting synthetic dataset transfers LoRA modules to new models. We show the effectiveness of our approach using both LLama and Gemma model families. Our approach achieves lossless (mostly improved) LoRA transfer between models within and across different base model families, and even between different PEFT methods, on a wide variety of tasks.

## 1 Introduction

The remarkable progress in language modeling has led to the development of Large Language Models (LLMs) [16, 13, 4, 2], achieving high performance on general language tasks via scaling model parameters to multi-billion sizes. Despite their great progress, even the largest and strongest LLMs [16] still significantly benefit from fine-tuning to downstream tasks for enhanced specialization and consequent performance improvement [50]. However, it is commonly difficult to gain the computational, memory, and disk resources needed for fine-tuning and later hosting fine-tuned large-scale models, especially when serving model customization APIs to numerous clients. Thus, a common approach to LLM finetuning is to use parameter-efficient finetuning (PEFT) methods, the most widespread of which are Low-Rank Adapters (LoRA) [31, 43], which only train a small number of additional parameters while freezing the base pre-trained model. Using PEFT can lead to more efficient and compute-friendly training without sacrificing final performance [31], as well as allowing efficient serving of large quantities of LoRA models 'orbiting' a common base model 'core' [56]. However, a LoRA model fine-tuned for a specific task is tied to its base model and cannot be used without it, and also cannot be directly transferred to another base model. This is quite problematic in commercial cloud model serving scenarios, where after the base

---

[*]MIT

[†]MIT-IBM Watson AI Lab

[‡]Work done while at MIT-IBM Watson AI Lab

38th Conference on Neural Information Processing Systems (NeurIPS 2024).

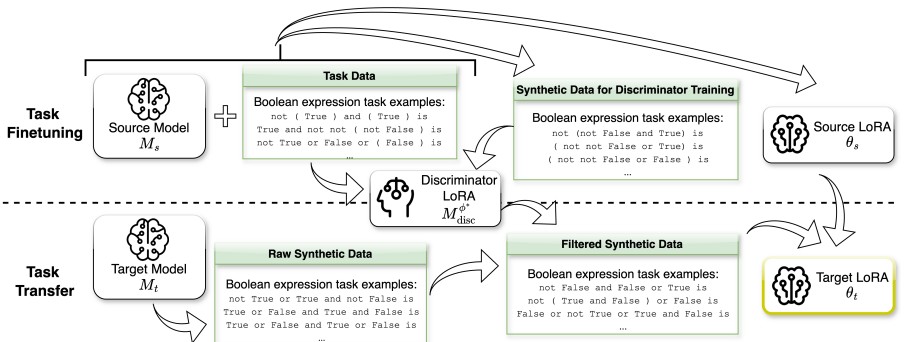

Figure 1: *Trans-LoRA* overview. Examples from 'boolean expressions' BBH task illustrate the lower diversity of raw synthetic samples compared to the original task data, which is fixed by our filtering approach. The source model is used to: 1. train the source LoRA; 2. synthesize data for discriminator training; and 3. train the (LoRA) discriminator. Then, the target model is used to synthesize data for transfer (filtered by discriminator) and train target LoRA using the source LoRA teacher.

model needs to be deprecated and replaced by a newer LLM, the (potentially thousands of) clients' LoRA models need to be switched to the new base model. Naively, one would have to re-train all the LoRA models, which, understandably, is a logistic nightmare given that clients' proprietary task data is commonly confidential and is not retained on the servers of the cloud service provider. Naturally, asking all of the clients to re-send the data for re-training or retraining on their own is neither scalable nor practical.

In this work, we propose *Trans-LoRA* - an approach for 'universal' LoRA transfer offering an ability to train LoRA models in a way that allows them to be transferred to new base models, and even to other kinds of PEFT (e.g. LoRA [31] ↔ DoRA [43] or PT[37]), in an automatic and centralized manner on the model service provider side, while preserving or improving performance, and without the need to access to the clients' data used to train the original LoRAs. Our *Trans-LoRA* is based on using the source base model LoRA to teach the target base model LoRA, while the main challenge is in obtaining the training curriculum for such a transfer in a manner that is both data-free and sufficiently effective to guarantee the resulting LoRA performance improvement beyond the maximum of the respective target base model and the source LoRA performances. Surprisingly, in *Trans-LoRA* we demonstrate it is possible to obtain an effective transfer curriculum for achieving these feats using synthetic data generated from the target base model. However, this by itself is insufficient to obtain the specified guarantees. We discover in *Trans-LoRA* that we need to additionally train a discriminator model for synthetic data filtering. Our proposed discriminator is trained on a mix of synthetic and real data alongside the source LoRA model and is optimized to ensure the filtered synthetic data most closely resembles the source LoRA training distribution. We provide extensive evidence, insights, and ablations as to why the proposed *Trans-LoRA* synthetic transfer curriculum works and is superior to the alternative curriculum-building approaches.

We perform numerous experiments confirming that our *Trans-LoRA* achieves the above guarantees while transferring within and across the popular Llama2 [15] and Gemma [14] model families, popular LoRA [31], DoRA [43], and Prompt Tuning [37] PEFT variants, and using a large variety of about 90 (language, code, and math) tasks contained in popular datasets such as BBH [60], MMLU [28], GSM8K [10], MBPP [5], and MBPP+ [41]. Notably, our *Trans-LoRA* not only achieves overall lossless transfer, it primarily improves performance (by up to $10\%$ in some cases) over the maximum among the fine-tuned source model and the target base model performances, thus consistently achieving positive transfer! We perform an ablation comparing to transfer using unfiltered synthetic data or random data from other sources. We explore transferring between different PEFT variants (e.g., LoRA[31] ↔ DoRA[43] or PT[37]), as well as multi-step transfer through an intermediate model (simulating multiple transfers due to consecutive model deprecations), in all cases supporting the robustness and merits of our *Trans-LoRA* approach. We also show that our *Trans-LoRA* positively benefits from scaling the synthetic data generation. Finally, we provide further error analysis of *Trans-LoRA* and ways to mitigate some edge-case scenarios.

To the best of our knowledge, *Trans-LoRA* is the first approach to explore the automatic, nearly data-free, and universal transferability of LoRA (or any other PEFT) models between base (LLM) models. The effectiveness of our approach observed in numerous experiments and ablations strongly suggests that our *Trans-LoRA* can be readily used for the said tasks in the challenging and yet very practical massive-scale custom models serving cloud applications.

## 2   Related Work

**Parameter Efficient Finetuning (PEFT)**   has emerged as an important area of research, particularly in the domain of transfer learning where the adaptation of large pre-trained models to specific tasks without extensive retraining is a significant challenge [17, 39, 25, 12, 70]. The literature on PEFT spans various approaches, each characterized by its strategy to modify a minimal number of parameters while maintaining competitive performance. Many different PEFT methods have been proposed, spanning Adapter Modules [30, 75, 26, 59], Prompt Tuning [37, 32] including multi-task variants [66], very popular Low-Rank Adaptation techniques including LoRA [31], DoRA [43, 69, 54], NOLA [36] and others [77, 33, 71]. A major challenge with PEFT techniques is that they do not transfer across base models and our proposed approach addresses this challenge for the first time.

**Knowledge Distillation (KD)**   is a technique where knowledge from a larger, typically more complex model (teacher) is transferred to a smaller, more efficient model (student) [29, 21, 35, 51, 53, 45]. Additional variants proposed include Self-Distillation [73, 3, 74, 47, 76] with same model as teacher and student, and Weak to Strong Distillation [6, 63, 34] that can under some circumstance help the stronger model to avoid overfitting [9]. While these approaches have shown promise in transferring between models, they still rely on training corpus for the distillation making them challenging to apply in a data-free scenario. We see from our experiments that producing a good set of data for distillation that would guarantee lossless transfer of PEFT models between base models and/or PEFT types is challenging and addressed by our proposed approach.

**Synthetic Data**   is increasingly used to train machine learning models [7, 49, 1, 52]. It has been used in computer vision [20, 11, 24, 42, 62], language processing [23, 27, 64, 8, 55], and more recently in instruction tuning and LLM alignment [65, 61, 67, 48, 46, 38, 58]. While synthetic data has been researched for general model improvement, to the best of our knowledge we are the first to explore its use for PEFT models transfer between base models and PEFT variants. As we show in our experiments and ablations, lossless transfer can only be achieved with careful curation of synthetic data achieved in our approach in an automatic and nearly source-data-free way. Additionally, we highlight that the synthetic data filtering approach employed in *Trans-LoRA* can be orthogonally applied on top of any of the more advanced synthetic data generation methods [72, 19, 44, 68].

## 3   *Trans-LoRA*

Given a pre-trained model $\mathcal{M}_s$ (dubbed the source model going forward) and a task-specific dataset, $\mathcal{D} = \{\boldsymbol{x}_n, \boldsymbol{y}_n\}_{n=1}^N$ of prompt ($\boldsymbol{x}_n$) and completion ($\boldsymbol{y}_n$) pairs, we assume that we have tuned $\mathcal{M}_s$ on $\mathcal{D}$ using a PEFT method (e.g., LoRA [31]), obtaining a task-adapted set of *additional* parameters $\boldsymbol{\theta}_s$ (e.g., realized as a set of residual adapters in [31]). Next, given a distinct model $\mathcal{M}_t$ (the target model) and access to only a *small* subset of 'seed' examples, $\bar{\mathcal{D}} \subset \mathcal{D}$, our goal is to learn task-adapted parameters $\boldsymbol{\theta}_t$ for $\mathcal{M}_t$ such that $\boldsymbol{\theta}_t$ bestows similar or better capabilities on $\mathcal{M}_t$ as those bestowed by $\boldsymbol{\theta}_s$ on $\mathcal{M}_s$. In *Trans-LoRA* we consider $\bar{\mathcal{D}}$ to be a very small set of demonstrations ($|\bar{\mathcal{D}}| = 5$ in all experiments) explaining the intent of the task and its I/O format. Keeping this tiny set of 5 samples $\bar{\mathcal{D}}$ does not violate the nearly data-free property of our *Trans-LoRA*, as $\bar{\mathcal{D}}$ can be cleaned from proprietary information, retaining only the core expected properties of the task.

### 3.1   Capabilities transfer through knowledge distillation on synthetic data

While $\mathcal{D}$ is unavailable when training $\boldsymbol{\theta}_t$ for $\mathcal{M}_t$, we do have $\boldsymbol{\theta}_s$, $\mathcal{M}_s$, and $\bar{\mathcal{D}}$ available to us. As such, capabilities can be transferred between $\boldsymbol{\theta}_s$ and $\boldsymbol{\theta}_t$ via knowledge distillation, i.e., by tuning $\boldsymbol{\theta}_t$ to match the completions produced by $\mathcal{M}_s$ with the task-adapted parameters $\underline{\boldsymbol{\theta}}_s$. Unfortunately, naively distilling on $\bar{\mathcal{D}}$ performs increasingly poorly with shrinking cardinality of $\bar{\mathcal{D}}$ and is often detrimental to a point where the un-adapted $\mathcal{M}_t$ outperforms $\boldsymbol{\theta}_t$ tuned on $\bar{\mathcal{D}}$. This is particularly so for $|\bar{\mathcal{D}}| = 5$ (Section 4.3) set by us as a requirement for *Trans-LoRA* to maintain its appealing nearly data-free aspect.

But if $\bar{\mathcal{D}}$ is insufficient, and the original task data cannot be retained, what should then be used as the necessary input samples (outputs are not required) for the knowledge distillation? One attempt could just be using random pieces of text from the web (e.g. from Wikipedia). However, these samples do not follow the input distribution of the task and result in a poor transfer (Section 4.3). A key insight behind our approach is that augmenting $\bar{\mathcal{D}}$ with carefully synthesized data, $\mathcal{D}_{\text{syn}}$, allows for effective learning of $\boldsymbol{\theta}_t$. However, interestingly, naive synthesis (e.g. from $\mathcal{M}_t$) using $\bar{\mathcal{D}}$ as demonstrations is by itself *insufficient* (Section 4.3) to produce the set of inputs for *lossless* transfer, that is for guaranteeing $\mathcal{M}_t + \boldsymbol{\theta}_t$ outperforms both the

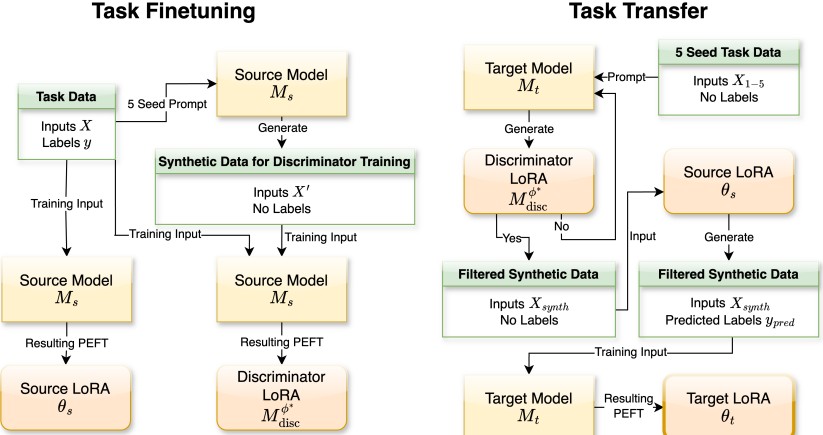

Figure 2: Detailed breakdown of *Trans-LoRA*. **Task Finetuning** is done beforehand and produces the source LoRA for the source model and the discriminator. **Task Transfer** utilizes the source LoRA and discriminator to transfer the LoRA onto the target model and produce the target LoRA.

non-tuned $\mathcal{M}_t$ and the $\mathcal{M}_s + \boldsymbol{\theta}_s$ as desired. We find that in addition to following the task distribution (which can be approximated via synthesizing from $\bar{\mathcal{D}}$ as demonstrations), the synthetic data must also adhere to one additional important requirement - it must also follow the distribution used to sample the original training set $\mathcal{D}$ out of all possible task data. Clearly, this marginal distribution $\mathcal{P}$ of just the inputs $\{x_n\}$ of the training samples in $\mathcal{D}$ would intuitively correspond to the 'comfort zone' of the intended teacher model $\mathcal{M}_s + \boldsymbol{\theta}_s$ (that learned from observing $\mathcal{D}$ and not the entire task data). Hence making it more likely for $\mathcal{M}_s + \boldsymbol{\theta}_s$ to produce higher quality outputs for the transfer for inputs sampled from $\mathcal{P}$.

Using above intuitions, we build a synthetic data simulator that generates data $\mathcal{D}_{\text{syn}}$ that is statistically indistinguishable from the observed task data $\mathcal{D}$ and is used for the aforementioned knowledge distillation at the time of transfer. Drawing inspiration from GAN [20], our simulator consists of a generator and a discriminator, described in greater detail below. While the generator part of the simulator is achieved by an LLM endowed with our designed prompt and using the tiny $\bar{\mathcal{D}}$ as in-context demonstrations, the discriminator is a separate PEFT model trained once alongside the training of $\boldsymbol{\theta}_s$ on $\mathcal{D}$ and kept for all future transfers. Hence we can safely assume access to $\mathcal{D}$ for discriminator training. Discriminator training does not require knowledge of the target model $\mathcal{M}_t$.

**Data synthesis via a large language model generator.** We use an instruction-tuned LLM $\mathcal{M}_{\text{gen}}$ and prompt it to generate prompt and completion pairs similar to those in $\bar{\mathcal{D}}$. In our experiments, we used the target model $\mathcal{M}_t$ itself for $\mathcal{M}_{\text{gen}}$, but any model capable of following detailed instructions can be used in its place. See Appendix A.1 for the prompt we used for data synthesis.

**Data filtration via a large language model discriminator.** To train a discriminator that would be able to effectively filter synthetic data, determining how close a synthetic sample is to the marginal distribution of the inputs in $\mathcal{D}$, we need a synthetic sample set. This synthetic sample set is to serve as 'negatives' for the discriminator training while the 'real' inputs from $\mathcal{D}$ serve as positives. As stated above, during subsequent transfers of the PEFT model to future models $\mathcal{M}_t$ we use these $\mathcal{M}_t$ models themselves for the synthetic data generator. However, we do not have access to them during the discriminator training (as it is trained in parallel to the source PEFT model). Hence, we use synthetic data generated from $\mathcal{M}_s$ for our discriminator training and surprisingly find that the resulting discriminator generalizes well to filter synthetic data for a variety of unseen downstream generators ($\mathcal{M}_t$) as evaluated in our experiments (Section 4). For our discriminator, we use an LLM, $\mathcal{M}_{\text{disc}}^{\phi}$, endowed with a small set of learnable parameters, $\phi$. We learn $\phi$ by optimizing,

$$\phi^* = \underset{\phi}{\text{argmax}} \ \mathbb{E}_{\boldsymbol{x} \sim D}[\log p_{\mathcal{M}_{\text{disc}}^{\phi}}(\text{``yes''} | \text{t}(\boldsymbol{x}))] + \mathbb{E}_{\boldsymbol{x} \sim \mathcal{M}_s}[\log p_{\mathcal{M}_{\text{disc}}^{\phi}}(\text{``no''} | \text{t}(\boldsymbol{x}))], \quad (1)$$

where, we use $\text{t}(\boldsymbol{x})$ to represent the prompt, "$\boldsymbol{x}$ \n Is the above question from NAME dataset?" and replace NAME with a short descriptor identifying the dataset from which $\mathcal{D}$ is drawn; and $\boldsymbol{x} \sim \mathcal{M}_s$ represents sampling from our synthetic data generation process for the task as explained above (with $\mathcal{M}_s$ as the generator LLM in this case). See Appendix A.1 for the specific prompts we used. In our experiments, we used the source model $\mathcal{M}_s$ and LoRA to instantiate $\mathcal{M}_{\text{disc}}^{\phi}$.

Table 1: BigBench-Hard (BBH) collection averaged zero-shot results. The accuracies listed are averages of all 27 tasks from this collection. Evaluated using LM-Eval Harness [18].

| Source Model | Target Model | Discriminator Model | Source Model LoRA Acc. | Target Model no LoRA Acc. | Ours |
|---|---|---|---|---|---|
| Llama-2-7b | Llama-2-13b | Llama-2-7b | 43.32 | 37.85 | 43.41 |
| Gemma-2b | Gemma-7b | Gemma-2b | 31.84 | 37.75 | 43.61 |
| Llama-2-7b | Gemma-7b | Gemma-2b | 43.32 | 37.75 | 45.41 |
| Llama-2-7b | Gemma-7b | Llama-2-7b | 43.32 | 37.75 | 44.12 |

**Curating** $\mathcal{D}_{\text{syn}}$. At the time of PEFT transfer, we create $\mathcal{D}_{\text{syn}}$ by filtering generations from $\mathcal{M}_{\text{gen}}$ with the trained discriminator, $\mathcal{M}_{\text{disc}}^{\phi^*}$. We incorporate $\boldsymbol{x} \sim \mathcal{M}_{\text{gen}}$ into $\mathcal{D}_{\text{syn}}$ if $\mathcal{M}_{\text{disc}}^{\phi^*}$ is unable to recognize $\boldsymbol{x}$ as a synthetic sample, i.e., $p_{\mathcal{M}_{\text{disc}}^{\phi^*}}(\text{"yes"}|\text{t}(\boldsymbol{x})) > p_{\mathcal{M}_{\text{disc}}^{\phi^*}}(\text{"no"}|\text{t}(\boldsymbol{x}))$. Otherwise we discard $\boldsymbol{x}$. We repeat this rejection sampling procedure till the cardinality of $\mathcal{D}_{\text{syn}}$ equals that of $\mathcal{D}$.

We summarize our overall *Trans-LoRA* algorithm in Algorithm 1 and Figure 2.

## 4 Experiments

---
**Algorithm 1** *Trans-LoRA*

---

**Require:** $\bar{\mathcal{D}}, \boldsymbol{\theta}_s, \mathcal{M}_t, \mathcal{M}_{\text{disc}}^{\phi^*}$
   $\mathcal{M}_{\text{gen}} \leftarrow \mathcal{M}_t$
   $\mathcal{D}_{\text{syn}} \leftarrow \varnothing$
   **while** $|\mathcal{D}_{\text{syn}}| < |D|$ **do**
      $s \leftarrow \text{generate}(\mathcal{M}_{\text{gen}}, \bar{\mathcal{D}})$
      **if** $\text{verify}(\mathcal{M}_{\text{disc}}^{\phi^*}, s)$ **then**
         $\mathcal{D}_{\text{syn}} \leftarrow \mathcal{D}_{\text{syn}} \cup \{s\}$
      **end if**
   **end while**
   Initialize $\boldsymbol{\theta}_t$
   $\mathcal{H} \leftarrow \text{CrossEntropyLoss}()$
   **while** $\boldsymbol{\theta}_t$ not converged **do**
      $\mathcal{L} \leftarrow \mathcal{H}(\boldsymbol{\theta}_t(\mathcal{D}_{\text{syn}}), \boldsymbol{\theta}_s(\mathcal{D}_{\text{syn}}))$
      $\boldsymbol{\theta}_t \leftarrow \text{update}(\boldsymbol{\theta}_t, \mathcal{L})$
   **end while**

---

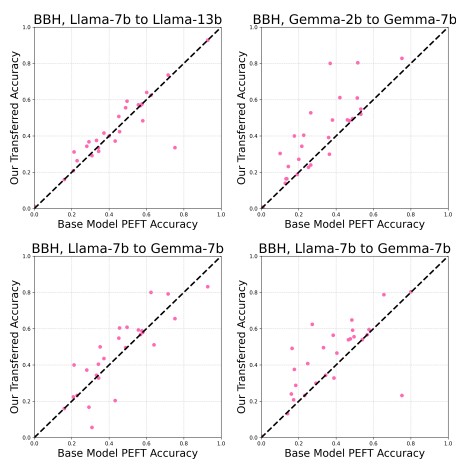

Figure 3: Transferred LoRA accuracy vs. source LoRA accuracy on BBH tasks. Details the rows of Table 1. Bottom left: row 3; Bottom right: row 4.

### 4.1 Experimental Setup

We have evaluated the effectiveness of our *Trans-LoRA* on two popular LLM families: Llama-2 [15] and Gemma [14], using 86 tasks from a large variety of topics from the following popular benchmarks: BigBench-Hard (BBH)[60] (27 reasoning tasks), Massive Multitask Language Understanding (MMLU)[28] (57 knowledge tasks), Mostly Basic Python Problems (MBPP)[5] (1 code task), and Grade School Math 8K (GSM8K)[10] (1 math task). BBH is a collection of 27 tasks where pre-existing LLMs could not outperform human evaluators. The tasks cover many different formats including multiple-choice, question answering, and short response. MMLU consists of 57 multiple-choice QA tasks testing common academic subjects with several difficulty levels. MBPP is a set of Python code generation problems with given problem descriptions and test cases. We also report results on MBPP+[41], which is built upon MBPP with more strict evaluations and added test cases. GSM8K dataset consists of a large number of grade school math problems. Due to the large number of training samples in GSM8K, we only pick the first 250 samples for fine-tuning our source LoRA models, and keep the number of filtered synthetic samples to 250 as in our other experiments.

More specifically, we attempted 4 groups of experiments of LoRA transfer on each collection of tasks: 1. transfer from Llama2-7b to Llama2-13b with Llama2-7b based discriminator; 2. transfer from Gemma-2b to Gemma-7b with Gemma-2b based discriminator; 3. transfer from Llama2-7b to Gemma-7b with Gemma-2b based discriminator; and 4. transfer from Llama2-7b to Gemma-7b with Llama2-7b based discriminator. We used the chat versions of Llama and the base versions of Gemma, thus exploring both

Table 2: Massive Multitask Language Understanding (MMLU) collection averaged zero-shot results. Accuracies are averages of all 57 tasks from this collection. Evaluated using LM-Eval Harness [18].

| Source Model | Target Model | Discriminator Model | Source Model LoRA Acc. | Target Model no LoRA Acc. | Ours |
|---|---|---|---|---|---|
| Llama-2-7b | Llama-2-13b | Llama-2-7b | 45.89 | 53.72 | 55.09 |
| Gemma-2b | Gemma-7b | Gemma-2b | 42.34 | 60.45 | 61.23 |
| Llama-2-7b | Gemma-7b | Gemma-2b | 45.89 | 60.45 | 61.12 |
| Llama-2-7b | Gemma-7b | Llama-2-7b | 45.89 | 60.45 | 61.22 |

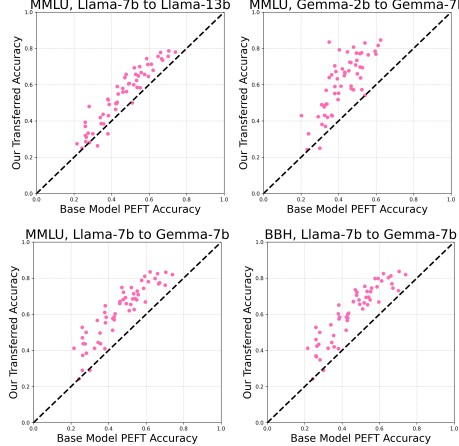

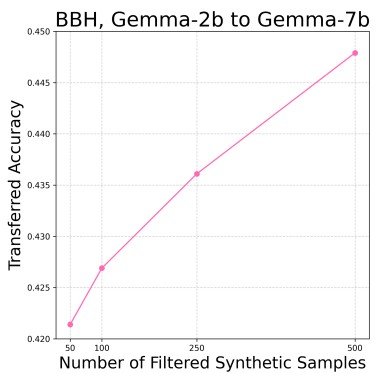

Figure 5: Scaling the number of synthetic samples generated through *Trans-LoRA*. Total training iterations in each experiment are kept identical for fair comparison. Done on BBH with Gemma-2b to Gemma-7b transfer and Gemma-2b as discriminator.

Figure 4: Transferred LoRA accuracy vs. source LoRA accuracy on MMLU tasks. Details the rows of Table 2. Bottom left: row 3; Bottom right: row 4.

within and across chat and base models LoRA transfer. We evaluate BBH, MMLU, and GSM8K using the Language Model Evaluation Harness [18], and we evaluate MBPP/MBPP+ using Evalplus [40]. We evaluate all our models under the zero-shot setting.

Hyperparameter-wise, we search the learning rate between $2*10^{-4}$ and $2*10^{-5}$ on the validation set using the AdamW optimizer with no weight decay and a linear learning rate scheduler without warmup. We end up adopting $2*10^{-4}$ for MMLU and $2*10^{-5}$ for all other tasks. We use a fixed 20 epochs for BBH, MBPP, and GSM8K and 10 epochs for MMLU. We train on the default LoRA configuration (adapters built only on query and value matrices of attention block) with effective batch size 8 (gradient accumulation used for larger models). We run on 1 V100 40GB GPU per transfer task. Each task takes on average 10 hours to finish. All tasks can be parallelized. [4]

### 4.2   Main Results

In Tables 1 to 4, we summarize the results for each task collection (BBH, MMLU, MBPP, and GSM8K respectively) for each source and target model combination. We test each task individually, and the results in each table are obtained by averaging over all the tasks in the respective collection. We observe that the LoRA models transferred by our *Trans-LoRA* consistently outperform both the source LoRAs and the target base models, demonstrating that our transfer is indeed *lossless*. Moreover, this suggests that our *Trans-LoRA* is effective at combining the information from LoRAs on a weaker source base model with the improved capabilities of a stronger target base model to create LoRAs on the target that are more powerful than both of them. And our *Trans-LoRA* is nearly data-free requiring almost no access to original tasks training data (beyond the 5 seed examples). We see that our *Trans-LoRA* consistently attains successful LoRA transfer independently of a specific combination of source, target, discriminator models, or the initial relative performance difference between the fine-tuned source LoRAs and the target base-models. We note that the performance increase of our transferred model is relatively smaller on MMLU compared to other tasks. As MMLU tasks are more knowledge-focused, we believe the pretraining is more influential

---

[4]Our code is provided in Supplementary and will be released upon acceptance. See Appendix A.3.

Table 3: Mostly Basic Python Problems (MBPP) zero-shot results. Presented in format of (standard MBPP evaluation / more strict MBPP+ evaluation). Evaluated using Evalplus [40].

| Source Model | Target Model | Discriminator Model | Source Model LoRA Acc. | Target Model no LoRA Acc. | Ours |
|---|---|---|---|---|---|
| Llama-2-7b | Llama-2-13b | Llama-2-7b | 27.2/25.0 | 37.1/31.7 | 39.7/34.4 |
| Gemma-2b | Gemma-7b | Gemma-2b | 41.1/33.9 | 37.9/32.1 | 50.0/40.6 |
| Llama-2-7b | Gemma-7b | Gemma-2b | 27.2/25.0 | 37.9/32.1 | 48.7/42.0 |
| Llama-2-7b | Gemma-7b | Llama-2-7b | 27.2/25.0 | 37.9/32.1 | 48.7/42.0 |

Table 4: Grade School Math 8K (GSM8K) no chain-of-thought prompting results.

| Source Model | Target Model | Discriminator Model | Source Model LoRA Acc. | Target Model no LoRA Acc. | Ours |
|---|---|---|---|---|---|
| Llama-2-7b | Llama-2-13b | Llama-2-7b | 19.64 | 28.86 | 30.70 |
| Gemma-2b | Gemma-7b | Gemma-2b | 14.94 | 40.64 | 44.58 |
| Llama-2-7b | Gemma-7b | Gemma-2b | 19.64 | 40.64 | 42.30 |
| Llama-2-7b | Gemma-7b | Llama-2-7b | 19.64 | 40.64 | 41.62 |

Table 5: Distillation curriculum ablations on 27 tasks of the BigBench-Hard (BBH) collection.

| Model Config | Source Model PEFT Acc. | Target Model no PEFT Acc. | Random Wikipedia | Unfiltered Synthetic Data | 5 Seed Samples | Ours |
|---|---|---|---|---|---|---|
| Source: Llama-2-7b Target: Llama-2-13b Discriminator: Llama-2-7b | 43.32 | 37.85 | 37.32 | 41.95 | 39.82 | 43.41 |

Table 6: *Trans-LoRA* for transferring between different base models *and* different PEFT methods on BigBench-Hard (BBH). Accuracies are zero-shot averaged results of all tasks from this collection.

| Source Model | Target Model | Discriminator Model | Source Model PEFT Acc. | Target Model no PEFT Acc. | Ours |
|---|---|---|---|---|---|
| Gemma-2b (LoRA) | Gemma-7b (LoRA) | Gemma-2b | 31.84 | 37.75 | 43.61 |
| Gemma-2b (LoRA) | Gemma-7b (DoRA) | Gemma-2b | 31.84 | 37.75 | 40.74 |
| Gemma-2b (DoRA) | Gemma-7b (LoRA) | Gemma-2b | 33.07 | 37.75 | 41.81 |
| Gemma-2b (DoRA) | Gemma-7b (DoRA) | Gemma-2b | 33.07 | 37.75 | 41.40 |
| Gemma-2b (LoRA) | Gemma-7b (PT) | Gemma-2b | 31.84 | 37.75 | 43.99 |
| Gemma-2b (PT) | Gemma-7b (LoRA) | Gemma-2b | 33.25 | 37.75 | 38.14 |
| Gemma-2b (PT) | Gemma-7b (PT) | Gemma-2b | 33.25 | 37.75 | 42.90 |

than the finetuning for MMLU. We also experimentally verified that increasing finetuning epochs (without adding more synthetic data) on MMLU does not lead to further improvements.

For more details, Figures 3 and 4 show a detailed distribution of LoRA transfer results for each task from the BBH and MMLU collections. We see that in both cases, the majority of data points are near or above the $y = x$ line (the dotted line), indicating our transferred target LoRAs match or outperform the source ones. These individual task distributions demonstrate the robustness of our *Trans-LoRA*. We analyze the few outliers in Section 5.

### 4.3 Ablation Experiments

**Distillation Data**   Here we evaluate the effect of the choice of the input data for distillation. As varying kinds of transfer on numerous tasks is time and resource-consuming, we run this ablation only on BBH tasks and the 'between Llama-2 models transfer' (most challenging, smallest gains) objective. Results are summarized in Table 5. We compare distilling the source LoRAs on: (1) random Wikipedia text; (2) raw

Table 7: Continuous transfer on several models on BigBench-Hard (BBH). We transfer from source model to intermediate model, then from intermediate model to target model, all using the same discriminator model. Accuracies are zero-shot averaged results of all tasks from this collection.

| Model Config | Source Model LoRA Acc. | Intermediate Model no LoRA Acc. | Our Transferred Intermediate Model | Target Model no LoRA Acc. | Our Transferred Target Model |
|---|---|---|---|---|---|
| Source: Llama-2-7b Intermediate: Llama-2-13b Target: Gemma-7b Discriminator: Llama-2-7b | 43.32 | 37.85 | 43.41 | 37.75 | 45.04 |

Table 8: Experiments on T5 models and 3 additional tasks, where our results are reported on *Trans-LoRA* transfer from T5-L finetuned LoRA to T5-XL base model.

| Dataset | T5-L Finetuned | T5-XL Base | Ours |
|---|---|---|---|
| Coqa | 32.60 | 55.84 | 61.44 |
| Newsroom | 85.09 | 84.19 | 85.70 |
| Squadv2 | 95.40 | 96.32 | 98.48 |

synthesized samples without discriminator filtering; (3) only the 5 seed samples used for data synthesis; and (4) our *Trans-LoRA*. From Table 5, we see that our *Trans-LoRA* outperforms other baselines by a large margin, indicating that: (a) synthetic data designed to mimic task data is highly beneficial, and random or seed data does not suffice; and (b) discriminator filtering is effective providing good gains over raw synthetic data. These results further verify our hypothesis on the importance of the proximity of distillation inputs to the original training data.

**Other PEFT Methods**    To further illustrate the robustness and wide applicability of our *Trans-LoRA*, we test its ability to transfer non-LoRA PEFT models. In particular, we apply our *Trans-LoRA* to Weight-Decomposed Low-Rank Adaptation (DoRA)[43], and Prompt Tuning (PT) [37]. For DoRA, we use the same set of hyperparameters as LoRA, and for Prompt Tuning we use a higher learning rate of $2*10^{-3}$ and initialization text provided in Appendix A.2. Table 6 indicates that despite the change of the specific PEFT approach, we can achieve satisfactory results upon transfer.

**Continuous Transfer**    To further verify the practical use-case of using our *Trans-LoRA* for several transfers in a row, we evaluate continuous transfer, where the LoRA model is transferred from source to target via an intermediate model. The discriminator model is kept the same throughout this process, closely mimicking real-world application scenarios where the discriminator model needs to be re-used for all subsequent transfers. From Table 7, we see that continuous transfer does not lead to degradation in performance. This result proves the robustness and practicality of our *Trans-LoRA*, where the client only needs to deliver the discriminator and trained PEFT once to allow for multiple future transfers to different future base models.

**Scaling the amount of Synthetic Samples**    Another advantage of our *Trans-LoRA* is the theoretically unlimited data synthesis process. In all previous experiments, we kept the number of filtered synthetic samples to be the same as the number of samples in the original training dataset (set to 250). We show in Figure 5 that our *Trans-LoRA* exhibits good scaling behavior w.r.t. the number of filtered samples generated, which gives the user the freedom to balance the trade-off between final task accuracy and total compute.

**Additional experiments**    To demonstrate the effectiveness of our approach on a wider range of tasks and models, we performed additional experiments on T5 series model and 3 additional tasks. The results are shown in Table 8.

Table 9: Maximum mean discrepancy(MMD) comparing filtered and unfiltered synthetic data with original dataset using first 4 tasks of BBH. Smaller values indicate smaller distance to original dataset.

| Task Name | Filtered Data MMD | Unfiltered Data MMD |
|---|---|---|
| boolean expressions | 0.7155 | 1.3072 |
| causal judgement | 0.2255 | 0.7714 |
| date understanding | 0.2438 | 0.8282 |
| disambiguation QA | 0.2097 | 0.9231 |

## 5 Analysis

### 5.1 Cost analysis

Our *Trans-LoRA* relies on training an additional discriminator. We summarize the empirical cost associated with this process and demonstrate that it only incurs a negligible overhead. Discriminator training typically takes just 1 epoch to reach over 90% accuracy. LoRAs were trained for 20 epochs in our experiments which was empirically observed to produce the best performance for the source LoRA models. Given that both discriminator training and LoRA training used equal number of samples per epoch (half real half synthetic for discriminator training, full real data for LoRA training), the cost of training discriminator is only around 1/20 of the training cost of LoRA modules. Synthetic data generation for training discriminators took less than 5 minutes for most tasks on a single V100 (for all base models). These costs are almost negligible compared to training the source LoRAs.

### 5.2 Distribution of filtered synthetic data

In order to provide a more direct understanding of the difference between filtered synthetic samples from our *Trans-LoRA* and unfiltered raw synthetic samples, we encode each sample into vector representation using a MPNet encoder [57] and calculate maximum mean discrepancy [22] on the encoded representations. The maximum mean discrepancy can be viewed as a measure of distance between two high dimensional distributions, or in other words how much of the original distribution can be explained by the given distribution. We run this analysis on the first 4 BBH tasks with synthetic data filtered by their respective Llama2-7b discriminators from the Llama2-7b to Llama2-13b LoRA transfer experiment. From Table 9, we clearly observe lower MMD values for our filtered synthetic data, confirming the utility of the discriminators employed in our *Trans-LoRA*.

To prove that the data we generate through *Trans-LoRA* is fundamentally different from the original data and the discriminator in *Trans-LoRA* does not simply memorize original samples, we performed further analysis on the us_foreign_policy task under MMLU. We find the closest pair of questions from the real data and our synthesized data under the embedding space of a pretrained MPNet. This closest pair has a Euclidean distance of 0.604, which indicates that there is absolutely no overlap between synthetic samples and real samples. This closest pair consists of: "What were the implications of the Cold War for American exceptionalism?" (real) and "What was the significance of the Cold War to the development of American foreign policy?" (synthesized). These questions are asking for completely different aspects of the subject. We also exhibit the T-SNE plot on the embeddings in Appendix (Figure 7). Although the distributions of synthetic data and real data are similar, they do not share any identical points.

### 5.3 Error Analysis

We see from Figures 3 and 4 that for very few of our 86 evaluated tasks, the performance of LoRAs transferred by our *Trans-LoRA* may become lower than the baseline. In this section, we take a closer look at one such specific example task: Disambiguation-QA from BBH to analyze why this occurred.

**Insufficient understanding of task**  Example comparison of problematic synthetic vs. real task data is in Figure 6. The generated synthetic question is not valid because none of the answers is correct. In this example, the generator model does not seem to have correctly understood the task intent; rather, it just mimicked the pattern of the real samples. We observe similar failed samples for (the few) other tasks residing under the Figures 3 and 4 diagonals.

**Solution**  We observe that increasing the number of real samples used to prompt the data synthesis (i.e., increasing $|\tilde{\mathcal{D}}|$) can effectively help the generator model to learn the inherent reasoning and structuring

```
In the following sentences,
explain the antecedent of the
pronoun (which thing the pro-
noun refers to), or state that
it is ambiguous.
Sentence: Everyone in the class
had to wear a uniform except
for Sarah, who had to wear
something else.
Options:
(A) The uniform
(B) Something else
(C) Ambiguous
```

```
In the following sentences,
explain the antecedent of the
pronoun (which thing the pro-
noun refers to), or state that
it is ambiguous.
Sentence: Alex tells us that
they could not meet.
Options:
(A) Alex could not meet
(B) We could not meet
(C) Ambiguous
```

Figure 6: Comparison of problematic synthetic sample (left) and real sample (right) from disambiguation-qa task.

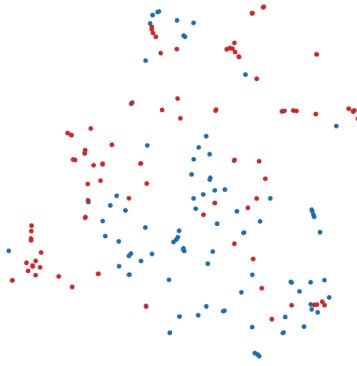

Figure 7: T-SNE plot of MPNet embeddings from us_foreign_policy (MMLU) dataset; Red points are our filtered synthetic data, blue points are real data.

of the task questions. Increasing the number of samples from 5 to 15 on disambiguation-qa, for example, leads to much more robust and realistic synthetic samples and significantly improved (up by 13%) performance. Thus, we recommend tuning the number of seed samples for synthesis when generated samples are not logically coherent and do not follow the task intent.

For more detailed analysis of our synthetic samples, we include a T-SNE plot of our samples under a pretrained embedding space in Figure 7.

# 6 Conclusions and Limitations

In this paper, we propose *Trans-LoRA*, an approach capable of nearly data-free LoRA model transfer between different base models (and even supporting transfer between different PEFT configurations) without requiring access to original task data. *Trans-LoRA* achieves equivalent or better performance when compared with the source LoRA and the target base model. To our knowledge, this paper is the first to explore the very practical use case of transferability of PEFT models. We hope that the success of our approach will inspire future explorations in this exciting research direction.

**Limitations** Our *Trans-LoRA* requires synthesizing data before the transfer requiring small, yet additional, compute. A promising future direction is to explore ways of direct PEFT transfer, without additional computation. Additionally, we discussed a potential limitation in task understanding by the synthesizer, observed in a few cases, and offered a path to mitigate it. We work with LLMs in our experiments, and although LLMs can sometimes produce harmful content, we rely on their authors for proper alignment.

## 7 Acknowledgements

This work was funded by MIT-IBM Watson AI Lab.

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

## A    Appendix

### A.1    Prompt Examples

Figure 8: Example prompt used in data synthesis for boolean expressions task from BBH.

```
Here are 10 examples:
1.  True and False or ( not True ) is
2.  not not True and not False or True is
3.  not False and False or False or False is
4.  True or False or not True or False is
5.  not not ( False and not False ) is
6.
```

Figure 9: Example prompt used in data filteration for boolean expressions task from BBH.

```
Answer in as few words as possible.
True and False or ( not True ) is
Is the above question from the boolean expressions
dataset?
```

## A.2 Prompt Tuning Initialization

Figure 10: Initialization for prompt tuning.

```
Answer the following question correctly:
```

## A.3 Code

Our code is available at https://github.com/raywang4/TransLoRA.

