# OpenReview forum: "$\textit{Trans-LoRA}$: towards data-free Transferable Parameter Efficient Finetuning"
_NeurIPS.cc/2024/Conference — NeurIPS 2024 poster_

### Official Review · Reviewer_7oKi · 2024-07-11

**Soundness:** 3
**Presentation:** 2
**Contribution:** 2
**Rating:** 5
**Confidence:** 4

**Summary:**

This paper proposes a nearly data-free method for transferring pre-tuned PEFT components (e.g., LoRA) between different models. To address issues of data inaccessibility, the authors propose to generate synthetic data from the target base model. To ensure that this synthetic data is in-distribution, they introduce an additional discriminator, which is trained concurrently with the source PEFT component.

**Strengths:**

1. The motivation is clear and interesting, focusing on compatibility issues between the base model and its PEFT components.
2. The writing is well-crafted and easy to understand.
3. The experimental design is robust, encompassing (i) compatibility within and across different base models, (ii) various PEFT methods, and (iii) a broad range of tasks.

**Weaknesses:**

1. The application scope of the proposed method appears limited due to the added constraints on training PEFT components: it necessitates training these components with an additional discriminator. This requirement is uncommon and incurs extra costs.
2. Further discussion on scalability is needed. As the number of PEFT components grows, updating all components seems time-consuming. By contrast, the approach in [1] suggests keeping all PEFT components static while only updating the base model in a specific way.
3. Additional properties of the synthetic data should be considered. While the proposed method focuses on in-distribution generation, the diversity of the synthetic data is also crucial.
4. More baselines of data-free knowledge distillation are needed, such as [2].
5. Given that the proposed method requires a small set of descriptive data, it may be more accurately described as "data-efficient" rather than "data-free."

[1] TaCA: Upgrading Your Visual Foundation Model with Task-agnostic Compatible Adapter, arXiv 2023.

[2] Prompting to Distill: Boosting Data-Free Knowledge Distillation via Reinforced Prompt, IJCAI 2022.

**Questions:**

See weaknesses.

**Limitations:**

The authors discuss several limitations:
(i) increased costs associated with data generation,
(ii) potential misunderstandings of the task.

---

> ### Author Rebuttal · Authors · 2024-08-06
>
> We would like to thank the reviewer for valuable feedback and comments on our paper. We appreciate the opportunity to address your concerns and clarify any misunderstandings. Below, we provide detailed responses to each of your comments.
>
> >The application scope of the proposed method appears limited due to the added constraints on training PEFT components: it necessitates training these components with an additional discriminator. This requirement is uncommon and incurs extra costs.
>
> Thank you for this suggestion, we are happy to elaborate on the discriminator overhead and would include this detail in the final version of the paper. Discriminator training typically takes just 1 epoch to reach over 90% accuracy. LoRAs were trained for 20 epochs in our experiments which was empirically observed to produce best performance for the source LoRA models. Given that both used equal amounts of samples per epoch (half real half synthetic for discriminator training), the cost of training discriminator is only around 1/20 of training cost of LoRA modules. Synthetic data generation for training discriminators took less than 5 minutes for most tasks on a single V100 (for all base models). These costs are almost negligible compared against training the original LoRA.
>
> > Further discussion on scalability is needed. As the number of PEFT components grows, updating all components seems time-consuming. By contrast, the approach in [1] suggests keeping all PEFT components static while only updating the base model in a specific way.
>
>
> We appreciate your suggestion. We are glad to cite Taca [1] in the related work, it is similar to our work in terms of the usage of distillation. However, Taca is in fact targetting one (very general) type of downstream task when being trained - the task of Vision-and-Language modeling - making image encoder tokens “readable” by an LLM decoder. If Taca is to be applied in our setting, it also needs to be trained separately for each downstream task. In this sense, our work is as scalable as Taca. Addressing the reviewer's concern on scalability, although each PEFT module needs to be transferred separately, these transfers can be executed independently in parallel to greatly speed up the process.
>
>
>
> >Additional properties of the synthetic data should be considered. While the proposed method focuses on in-distribution generation, the diversity of the synthetic data is also crucial.
>
>
> Thank you for pointing this out. As shown in an example T-SNE plot for the us_foreign_policy task (part of MMLU), the distributions of synthetic data and real data are just as diverse. They have similar coverages and characteristics as is apparent from the T-SNE plot. We will include this plot in our final paper.
>
>
> > More baselines of data-free knowledge distillation are needed, such as [2].
>
>
> Thank you for providing this reference! In fact, the baseline of data-free knowledge distillation is equivalent to the "unfiltered synthetic data" column in our ablation (Table 5), where samples are generated by simple prompting. We fixed our synthesis prompt to a simple prompt throughout all of our experiments for consistency and simplicity. Our primary contribution is the methodology allowing the use of synthetic data for lossless LoRA transfer. We believe that our approach’s ability to obtain consistent positive transfer results in all experiments using just the simple prompts underline its effectiveness. We also believe that other synthetic generation methods such as [2] are orthogonal to our approach. Any good data synthesis method can be applied together with our proposed Trans-LoRA approach. Combining these methods is a promising research direction for future work. We are happy to cite this work and include this discussion in the final paper.
>
> >Given that the proposed method requires a small set of descriptive data, it may be more accurately described as "data-efficient" rather than "data-free."
>
> We thank the reviewer for noting this, and we fully agree. We actually used the term “nearly data-free” in all 8 descriptions in our main paper except only 1 place in line 46. We will fix the reference on line 46 in our final version. Thanks!

---

> > ### Author Response · Authors · 2024-08-10
> >
> > Dear Reviewer 7oKi,
> >
> > We sincerely appreciate your valuable feedback and the time you've taken to review our submission. Please let us know if our response has satisfactorily addressed your concerns, and we are more than willing to address any additional comments you may have.
> >
> > Thank you!

---

> > ### Comment · Reviewer_7oKi · 2024-08-12
> >
> > Thank you for the author's rebuttal, which has addressed most of my concerns. I still have the following points to discuss with the author:
> >
> > 1. Even if training a discriminator is not costly, users will not train an additional discriminator, as it does not aid their fine-tuning tasks. However, the method proposed by the author makes an additional assumption about user behavior, assuming that users train a LoRA while also training a discriminator, which is not the case in practice. Therefore, I say the application of this method is limited.
> >
> > 2. Reference [2] is not merely "unfiltered synthetic data." It employs model inversion techniques to invert a topic prompt, which is then fed into a pre-trained language model to produce in-distribution data, similar to your objective.
> >
> > 3. Model inversion is also a technique used to generate data by optimizing inputs/prompts so as to produce a target output. Could you compare model inversion and prompting generation?
> >
> > [2] Prompting to Distill: Boosting Data-Free Knowledge Distillation via Reinforced Prompt, IJCAI 2022.

---

> > > ### Author Response · Authors · 2024-08-13
> > >
> > > Thank you for your continued engagement and for providing further insights. We appreciate the opportunity to clarify and discuss the points you have raised.
> > >
> > > >Even if training a discriminator is not costly, users will not train an additional discriminator, as it does not aid their fine-tuning tasks. However, the method proposed by the author makes an additional assumption about user behavior, assuming that users train a LoRA while also training a discriminator, which is not the case in practice. Therefore, I say the application of this method is limited.
> > >
> > > To clarify, our approach in fact simplifies the process for user and enables an effortless update of the user's model from the user's perspective.
> > >
> > > One desired application setting of our approach is when a user provides their private data to a service provider for training and hosting a PEFT model. The service provider will train both the LoRA for task data and LoRA for discriminator and then delete the private data. There are no additional effort required on the user's side. When the base model needs to be deprecated and updated by a newer model, the provider can simply use the prepared discriminator for transferring the PEFT model without asking the client again for their private data. Thus, our approach simplifies the process for the user, as they now only need to provide their data once and be settled for all future updates (the discriminator only needs to be trained once even for multiple transfers).
> > >
> > > A user could also train a PEFT model and discriminator on their side to avoid providing sensitive data, which, given the low overhead of the discriminator, is quite practical.
> > >
> > > We acknowledge that our method is intended at all future users of PEFT models and will not apply to PEFT models prior to our method adoption. However, we believe it is a small limitation given the demonstrated success of our approach. We are happy to mention this in the limitation section. Thanks for suggesting!
> > >
> > > >Reference [2] is not merely "unfiltered synthetic data." It employs model inversion techniques to invert a topic prompt, which is then fed into a pre-trained language model to produce in-distribution data, similar to your objective.
> > >
> > > We used the term "unfiltered synthetic data" to refer that our approach and the approach in [2] are orthogonal and can be applied together. Our approach focuses on filtering of generated data, while the approach in [2] focuses on the generation itself. A straightforward combined approach is to use [2] to perform synthesis, and then apply our discriminator. We will cite [2] in our final paper and address this combination as a promising future direction.
> > >
> > > >Model inversion is also a technique used to generate data by optimizing inputs/prompts so as to produce a target output. Could you compare model inversion and prompting generation?
> > >
> > > Thank you for mentioning this topic. Model inversions are typically expensive and their operation requires backward passes for the inversion. For example, [2] trains a prompt generator alongside the student model using RL to guide the training of the prompt generator to prompts where student and teacher disagree. So each transfer would require such training and this might be prohibitively expensive. Our discriminator is one time and low cost, and any synthesis used in our method is forward only thanks to discriminator ability to filter useful samples.
> > >
> > > Additionally, model inversion is orthogonal to our discriminator filtering approach as well, similar to [2]. One can easily use model inversion for initial generation, then apply our discriminator filtering on the generated samples. We will address this in the discussion of our final paper.

---

> > > > ### Comment · Reviewer_7oKi · 2024-08-13
> > > >
> > > > Thank you again for the response. I will raise my score.

---

> > > > > ### Author Response · Authors · 2024-08-13
> > > > >
> > > > > Thank you so much for your continued support and for raising the score!
> > > > > We will make sure to incorporate all the points raised in the above discussion into the final version of the paper.

---

> ### Comment · Area_Chair_QhcS · 2024-08-12
>
> Dear Reviewer 7oKi,
>
> The authors have provided a rebuttal. Can you please provide your feedback after reading the rebuttal? The deadline is approaching fast.
>
> Thanks,
> AC

---

### Official Review · Reviewer_K48P · 2024-07-11

**Soundness:** 3
**Presentation:** 3
**Contribution:** 3
**Rating:** 6
**Confidence:** 4

**Summary:**

This paper proproses Trans-LoRA, a method that utilize synthetic data to transfer abilities learned using LoRA across different downstream tasks. Trans-LoRA first uses the source model for synthetic data generation. The generated synthetic data is used to train discriminator LoRA for filtering synthetic data for target model. Using filtered synthetic data together with source LoRA, the target model acquire synthetic data for training.
Trans-LoRA is validated on various benchmarks and models, showing the effacy of the method.

**Strengths:**

* The paper is clearly written and easy to follow
* Experiments are conducted on multiple benchmarks and different variants of LLaMA-2/Gemma
* Experiments show clear improvements in performance on all benchmarks.

**Weaknesses:**

* I understand that the current trend is to apply PEFT on decoder-only LLMs like LLaMa, but I'm a bit worried that the method might not be generic enough. I think the method might be suitable to alternative model architectures, for example
    * Encoder-only LMs like DeBERTa or RoBERTa. Since Trans-LoRA requires generation from the source model, it could be challenging to implement. One could work around this by using another LM with a decoder for generation/discrimination, but this approach seems suboptimal.
    * Similarly, ViT from the vision community.
* Also, the method seems to rely heavily on the generative ability of the source model/discriminator mdoel. I wonder if it's feasible to use weaker LMs like T5 or GPT2. The results in Table 5 also show that if we use random wikipedia (similarly to use a weaker LM), the performance drops significantly.
* One of the nice property of the method is one only needs 5 real samples to conduct PEFT. But computation-wise, one still needs a relatively large LM to generate samples/discriminate. Is that actually favorable in practice?
* minor: line 198: is it A100 40GB?

**Questions:**

See Weaknesses

---

> ### Author Rebuttal · Authors · 2024-08-06
>
> We would like to thank the reviewer for valuable feedback and comments on our paper. We appreciate the opportunity to address your concerns and clarify any misunderstandings. Below, we provide detailed responses to each of your comments.
>
> > I understand that the current trend is to apply PEFT on decoder-only LLMs like LLaMa, but I'm a bit worried that the method might not be generic enough. I think the method might be suitable to alternative model architectures, for example
>     -   Encoder-only LMs like DeBERTa or RoBERTa. Since Trans-LoRA requires generation from the source model, it could be challenging to implement. One could work around this by using another LM with a decoder for generation/discrimination, but this approach seems suboptimal.
>     -   Similarly, ViT from the vision community.
>
> We thank the reviewer for this suggestion. To better illustrate the generality of our Trans-LoRA approach we include the following experiment on the popular T5 model with Coqa, Newsroom, and Squadv2 datasets. The results included in the table below illustrate that our approach can be also effectively applied to generative models beyond “decoder only” (encoder-decoder in this case), as well as to weaker models than LLaMA and Gemma. We will include this result in the final manuscript. The paper is initially intended for decoder-only architectures, but exploring encoder-only models could be an interesting future direction.
> | Dataset | T5-L Finetuned | T5-XL Base | Ours |
> |:-:|:-:|:-:|:-:|
> | Coqa | 32.60 | 55.84 | 61.44|
> | Newsroom |85.09|84.19|85.70|
> | Squadv2 | 95.40|96.32|98.48|
>
> >Also, the method seems to rely heavily on the generative ability of the source model/discriminator mdoel. I wonder if it's feasible to use weaker LMs like T5 or GPT2. The results in Table 5 also show that if we use random wikipedia (similarly to use a weaker LM), the performance drops significantly.
>
> Thanks for the suggestion! In the above table, we tested with the T5 series model and demonstrated that our approach can be applied to these weaker language models as well on 3 separate tasks. We will include this result in the final manuscript. The experiment with transferring using random wikipedia data in Table 5 is only intended to show that data for the transfer is very important, it is not mimicking data generated from weaker models (as demonstrated by results on T5).
>
> >One of the nice property of the method is one only needs 5 real samples to conduct PEFT. But computation-wise, one still needs a relatively large LM to generate samples/discriminate. Is that actually favorable in practice?
>
> Thank you for this suggestion, we are happy to elaborate on the discriminator overhead and would include this detail in the final version of the paper. Discriminator training typically takes just 1 epoch to reach over 90% accuracy. LoRAs were trained for 20 epochs in our experiments which was empirically observed to produce best performance for the source LoRA models. Given that both used equal amounts of samples per epoch (half real half synthetic for discriminator training), the cost of training discriminator is only around 1/20 of training cost of LoRA modules. Synthetic data generation for training discriminators took less than 5 minutes for most tasks on a single V100 for all base models. These costs are almost negligible compared to training the original LoRA.
>
> In terms of synthesis during LoRA transfer, the generation process with the discriminator is quite fast (typically taking less than an hour for an entire dataset on V100). Moreover, our approach supports:
>
> 1. re-use of previously synthesized dataset (see Table 7)
>
> 2. parallelization of synthesis by using multiple GPUs.
>
> This implies that our Trans-LoRA approach is effective even when we only synthesize the dataset ONCE for all future transfers, and this one time synthesis can be done fast. Additionally, we also showed above that it is possible to use a smaller LM (T5) for our approach, which would greatly speed up generation as well.
>
> >-   minor: line 198: is it A100 40GB?
>
> Thank you for noting this! It means a V100 with 40GB of machine memory, rather than 40GB of GPU memory. We will make sure to clarify this in our final paper.

---

> > ### Comment · Reviewer_K48P · 2024-08-10
> >
> > Thank you for the detailed response. I'm glad to see the results on T5 and the explanation regarding computational cost.
> >
> > * Regarding W1, I may not have made my concern clear initially. My main issue is that the method may not be generic enough to work with encoder-only models. While the method works for decoder-only models, and thus naturally for encoder-decoder models, I can accept that it is specifically designed for LMs with decoders.
> > * I'm a bit confused about the settings used for T5. Could you please elaborate more on the details, such as the source, target, and discriminator?

---

> > > ### Author Response · Authors · 2024-08-10
> > >
> > > Thank you for the feedback!
> > > > Regarding W1, I may not have made my concern clear initially. My main issue is that the method may not be generic enough to work with encoder-only models. While the method works for decoder-only models, and thus naturally for encoder-decoder models, I can accept that it is specifically designed for LMs with decoders.
> > >
> > > Thank you for clarifying this! Since our paper is initially targeted at the most popular decoder-only models, we would consider applying on encoder-only models as a promising potential future direction. Based on the positive results on T5, it is likely that when applied on encoder-only models our approach still exhibits good performance. We will mention this point in our final paper.
> > >
> > > >I'm a bit confused about the settings used for T5. Could you please elaborate more on the details, such as the source, target, and discriminator?
> > >
> > > We are more than happy to elaborate: we used the most basic setting (row 1 and 2) in the main tables (Tables 1-4), where a T5-large model is used to train source LoRA and discriminator LoRA and the T5-XL model is used to distill the target LoRA. The results we reported are source model finetuned accuracy, target model base accuracy, and target model transferred accuracy (ours).

---

> > > > ### Author Response · Authors · 2024-08-12
> > > >
> > > > We thank again the reviewer for the support of our work and the constructive feedback! In case we have addressed all the reviewer’s concerns, we would appreciate if the reviewer would possibly consider improving the initial score. Thanks!

---

> > > > > ### Comment · Reviewer_K48P · 2024-08-12
> > > > >
> > > > > Thank you again for the response. After reading the new results and the authors' responses to other reviewers, I find myself between a borderline and weak accept decision. Unfortunately, within the limited timeframe for rebuttal, we didn't see results on more variations in base models and tasks, which would have provided a better understanding of the method's behavior. However, the method appears promising, especially with the new results on T5. I encourage the authors to explore this direction further, and I will increase my score to a weak acceptance.

---

> ### Author Response · Authors · 2024-08-13
>
> Thanks for your continued support and for raising the score!
> We will make sure to incorporate all the points raised in the above discussion into the final version of the paper.
> We agree that our approach obtains promising results and opens many interesting future work directions for us and others to explore further.

---

### Official Review · Reviewer_qZWK · 2024-07-12

**Soundness:** 2
**Presentation:** 3
**Contribution:** 2
**Rating:** 5
**Confidence:** 4

**Summary:**

The paper presents Trans-LoRA, a novel approach for transferring Low-Rank Adapter (LoRA) parameters across different base models without requiring access to the original task data. Trans-LoRA utilizes synthetic data generation and a discriminator model to filter this synthetic data, ensuring that the transferred LoRA parameters maintain or improve performance. The effectiveness of the method is validated through experiments on LLaMA and Gemma model families, demonstrating its utility across various tasks.

**Strengths:**

-The paper introduces a novel solution for transferring LoRA parameters across different base models without needing the original task data.
-The combination of synthetic data generation and discriminator filtering is innovative and well-suited for Parameter-Efficient Fine-Tuning (PEFT).
-The paper is well-organized and clearly written, with comprehensive explanations of the problem, proposed solution, and experimental setup.

**Weaknesses:**

-While the paper demonstrates the effectiveness of Trans-LoRA on LLaMA and Gemma models, evaluations on a wider range of models and tasks would strengthen its generalizability.
-The reliance on a discriminator introduces an overhead, and the paper does not fully address who will bear this cost in a cloud scenario.
- The paper relies on synthetic data generation, which can introduce biases and limitations. A more detailed discussion on these aspects would strengthen the paper.
- Exploring the potential integration of other synthetic data generation methods like ProGen: Progressive Zero-shot Dataset Generation via In-context Feedback and GOLD (Generalized Knowledge Distillation via Out-of-Distribution-Guided Language Data Generation) could enhance the data quality and effectiveness.
- Although the synthetic data generation process and discriminator filtering are designed to minimize risk, the potential for synthetic data to inadvertently reflect proprietary information remains a concern. Further safeguards and monitoring practices should be discussed to ensure data privacy.

**Questions:**

See weakness

**Limitations:**

The authors address several limitations, including the need for synthetic data generation and the potential computational overhead. However, the discussion could be expanded to include potential biases in synthetic data, variability of performance across different tasks, and the cost implications of the discriminator overhead in a cloud scenario. Additionally, more detail on the safeguards against data privacy risks would be beneficial.

---

> ### Author Rebuttal · Authors · 2024-08-06
>
> We would like to thank the reviewer for valuable feedback and comments on our paper. We appreciate the opportunity to address your concerns and clarify any misunderstandings. Below, we provide detailed responses to each of your comments.
>
> >While the paper demonstrates the effectiveness of Trans-LoRA on LLaMA and Gemma models, evaluations on a wider range of models and tasks would strengthen its generalizability.
>
> We thank the reviewer for this suggestion. To better illustrate the generality of our Trans-LoRA approach we include the following experiment on the popular T5 model with Coqa, Newsroom, and Squadv2 datasets. The results included in the table below illustrate that our approach can be also effectively applied to more tasks and models. We will include this result in the final manuscript.
> | Dataset | T5-L Finetuned | T5-XL Base | Ours |
> |:-:|:-:|:-:|:-:|
> | Coqa | 32.60 | 55.84 | 61.44|
> | Newsroom |85.09|84.19|85.70|
> | Squadv2 | 95.40|96.32|98.48|
>
> >The reliance on a discriminator introduces an overhead, and the paper does not fully address who will bear this cost in a cloud scenario.
>
> Thank you for this suggestion, we are happy to elaborate on the discriminator training overhead and would include this detail in the final version of the paper. Discriminator training typically takes just 1 epoch to reach over 90% accuracy. LoRAs were trained for 20 epochs in our experiments which was empirically observed to produce best performance for the source LoRA models. Given that both used equal amounts of samples per epoch (half real half synthetic for discriminator training), the cost of training discriminator is only around 1/20 of training cost of LoRA modules. Synthetic data generation for training discriminators took less than 5 minutes for most tasks on a single V100 (for all base models). These costs are almost negligible compared to training the original LoRAs.
>
> > The paper relies on synthetic data generation, which can introduce biases and limitations. A more detailed discussion on these aspects would strengthen the paper.
>
> Thanks for this suggestion! However, we have already provided a detailed MMD analysis of the synthetic data distribution in section 5.1, also ablating how the discriminator-based filtering improves on the MMD with the real data distribution. Moreover, we also highlighted some potential failures of synthetic data generation and even provided a potential mitigation addressing such failures in Section 5.2. We will further emphasize the importance of these aspects refering to these sections from the method section.
>
> >Exploring the potential integration of other synthetic data generation methods like ProGen: Progressive Zero-shot Dataset Generation via In-context Feedback and GOLD (Generalized Knowledge Distillation via Out-of-Distribution-Guided Language Data Generation) could enhance the data quality and effectiveness.
>
> Our primary contribution is proposing a methodology (Trans-LoRA) that demonstrates the possibility of using synthetic data for lossless LoRA transfer. Currently our experiments used a simple and straightforward prompting approach, while still demonstrating lossless and even positive transfer in all cases, further highlighting the effectivity of our Trans-LoRA approach. We believe that other synthetic generation methods, such as the great methods referenced by the reviewer, are orthogonal to our approach. Any good generation methods can be applied together with our Trans-LoRA, which is a promising direction left for future work to explore. We will add this to the discussion section of the final version of the paper, also citing the works mentioned by the reviewer (that are certainly applicable for this orthogonal future work extension).
>
> > Although the synthetic data generation process and discriminator filtering are designed to minimize risk, the potential for synthetic data to inadvertently reflect proprietary information remains a concern. Further safeguards and monitoring practices should be discussed to ensure data privacy.
>
> We performed further analysis on the us_foreign_policy task under MMLU. We find the closest pair of questions from the real data and our synthesized data under embedding space of a pretrained MPNet. This closest pair has a Euclidean distance of 0.604 (so there is absolutely no overlap between synthetic samples and real samples) and consists of: “What were the implications of the Cold War for American exceptionalism?” (real) and “What was the significance of the Cold War to the development of American foreign policy?” (synthesized). Note that these questions are asking for completely different aspects of the subject. We also exhibit the T-SNE plot on the embeddings in the shared response PDF. Although the distributions of synthetic data and real data are similar, they do not share any identical points. We will add this analysis and discussion to the final version of the paper, thanks for suggesting!

---

> > ### Author Response · Authors · 2024-08-10
> >
> > Dear Reviewer qZWK,
> >
> > We sincerely appreciate your valuable feedback and the time you've taken to review our submission. Please let us know if our response has satisfactorily addressed your concerns, and we are more than willing to address any additional comments you may have.
> >
> > Thank you!

---

> > > ### Comment · Reviewer_qZWK · 2024-08-12
> > >
> > > Thanks for replying to my questions and concerns. I have increased my score.

---

> ### Author Response · Authors · 2024-08-13
>
> We thank the reviewer for their continued support and for raising the score!
> We will make sure to incorporate all the points raised in the above discussion into the final version of the paper.

---

### Author Rebuttal · Authors · 2024-08-06

We would like to thank all the reviewers for reviewing our paper and providing valuable and constructive feedback.
We are grateful that the reviewers have highlighted our work as:

- well motivated (7oKi)
- innovative and novel in approach (qZWK)
- well-written and clear to understand (qZWK, K48P, 7oKi)
- experimentally solid and robust (qZWK, K48P, 7oKi)

We include supplementary materials in response to some of the reviewers' questions in the attached PDF.
To summarize the major responses we have made in rebuttal:

- We report the result of our approach on new models and datasets to demonstrate our applicability in weaker/smaller language models, encoder-decoder language models, and diverse model families.
- We address the questions on discriminator-related overheads, showcasing that our method only requires very little overhead cost.
- We discuss related works listed by reviewers and indicate that they are either orthogonal or complementary to our work.
- We include more evidence on the diversity and secureness of our filtered synthetic data, and provide specific quantitative examples to support the claims.

Again, we genuinely appreciate the input from reviewers and we thank all reviewers for their time and effort.

---

### Decision · Program_Chairs · 2024-09-25

**Decision:**

Accept (poster)

**Comment:**

The rebuttal addressed most of the concerns raised by the reviewers and now all reviewers support accepting the paper. Hence, it will be accepted. The paper presents a novel approach for transferring LoRA parameters across various base models. The idea is empirically validated by several experiments. Please